# Investigating the relationship between changes in social security benefits and mental health: a protocol for a systematic review

Julija Simpson [ID] , Heather Brown, Zoe Bell, Viviana Albani, Clare Bambra

Population Health Sciences Institute, Newcastle University, Newcastle upon Tyne, UK

**Correspondence to**
Julija Simpson;
julija.stoniute@newcastle.ac.uk

## ABSTRACT

**Introduction** Poor mental health is one of the greatest causes of disability in the world. Evidence increasingly shows that population mental health may be influenced by national social security policies. This systematic review aims to establish the relationship between social security and mental health in order to help inform recommendations for policy-makers, practitioners and future research.

**Methods and analysis** A systematic review of quantitative observational studies investigating mental health outcomes related to changes in social security policies will be conducted. Six major databases, including Medline, PsychInfo, Embase, Cumulative Index to Nursing and Allied Health Literature, Applied Social Sciences Index Abstracts and Scopus, as well as Research Papers in Economics will be searched from January 1979 to April 2020. The electronic database searches will be supplemented by reference and citation searches as well as hand-searching of key journals. The outcomes of interest are objective or subjective mental health outcomes, including stress, anxiety, depression, self-reported mental health scores, subjective well-being and suicide. Study selection will follow the Preferred Reporting Items for Systematic Reviews and Meta-Analyses guidelines, and the quality of the studies will be assessed by the validity assessment framework designed for appraising econometric studies. A narrative synthesis will be conducted for all included studies. If data permit, study findings will be synthesised by conducting a meta-analysis.

**Ethics and dissemination** As it will be a systematic review, without primary data collection, there will be no requirement for ethical approval. Findings will be disseminated through peer-reviewed publications and in various media, for example, conferences or symposia.

**PROSPERO registration number** CRD42019154733.

## INTRODUCTION

Mental illness is a major cause of mortality and morbidity worldwide.[1] In Organisation for Economic Cooperation and Development (OECD) countries, mental ill health affects an estimated 20% of the working-age population at any given time, causing major losses to individuals, communities and the economy.[2]

## Strengths and limitations of this study

► We employ a rigorous international gold-standard methodology of reporting Preferred Reporting Items for Systematic Reviews and Meta-Analyses Protocols (PRISMA-P) to facilitate the development and reporting of this protocol.

► By employing a PRISMA-P guideline, this study follows a clear framework to improve the transparency, accuracy and completeness of the systematic review protocol.

► To ensure comprehensiveness of the evidence base in the review, we have developed a broad search strategy, supplemented with reference and citation searches as well as hand-searching of key journals.

► A potential limitation of this study is that studies may be too heterogeneous to combine effect estimates.

It has been estimated that the global economy loses about US$1 trillion every year in productivity as a result of depression and anxiety.[3]

Evidence increasingly shows that mental ill health is socially determined.[4 5] As defined by the World Health Organisation, the social determinants include 'the conditions in which people are born, live, work and age, and the systems put in place to deal with illness. These circumstances are in turn shaped by a wider set of forces: economics, social policies and politics'[6].

One of the key policies affecting household income and thus living conditions—particularly those in the lowest socioeconomic groups—is social security policy. Broadly defined, social security policy includes income transfers from the state to individuals, with the aim to cover income loss due to poor health and disability, unemployment, low income, single parenthood and old age.[7]

A growing body of literature indicates that, although not explicitly designed, social security policies can have an impact on population mental health. For example, suicide

BMJ

rates in men in the USA fall sharply when individuals become eligible for social security retirement benefits, even though only a small fraction of men retire at this age.[8] Moreover, two major systematic reviews investigating the relationship between income and health in OECD countries also found that additional financial resources, including those from social security benefits, can have positive effects on both adult[9] and child mental health.[10 11] Lack of social protection, on the other hand, has been associated with negative effects on mental health. In the UK, the austerity-driven changes to the social security system have been associated with an increased prevalence of depression,[12] suicide[13] and mental health disorders.[14]

Although there is a growing body of literature documenting strong links between social disadvantage and mental health problems,[15 16] the evidence base on the health impact of social security reform remains small. A recent umbrella review of social protection policies in OECD countries has identified only two low-quality systematic reviews related to income maintenance policies.[17] Both of these reviews focused on specific policies (unemployment benefits and tax credits) and, to date, the broader range of social security policies and their impacts on mental health has not been reviewed. Furthermore, even though there have been broad systematic reviews of conditional cash transfers in low-income and middle-income countries,[18 19] to our knowledge, none of these reviews included mental health as an outcome. Therefore, in the context of global mental health crisis as well as rapidly rising concerns about the impact of recent austerity measures on population health and well-being, there is a growing need to better understand the relationship between social security reform and mental health.

The aim of this systematic review is to evaluate the impact of social security reform on mental health and mental health inequalities in high-income countries. By providing robust and concrete evidence, this review will add to the growing evidence base linking social determinants and mental health and will help inform recommendations for policy-makers, researchers and areas for further research.

## METHODS

The review protocol is reported in accordance with the standards of the Preferred Reporting Items for Systematic Reviews and Meta-Analysis Protocols[20] (PRISMA-P, see the online supplementary file 1).

Preliminary searchers were carried out in October 2019 and the study registered with the PROSPERO International Prospective Register of Systematic Reviews[21] on 31 October 2019. The review is planned to commence February 2020 and is anticipated to take 9 months to complete.

### Systematic review questions

1. What is the impact of social security reform on mental health?

2. What is the impact of social security reform on mental health inequalities?
   Mental health inequalities are defined as differences in mental health effects by commonly used indicators of socioeconomic status (SES) such as individual income, wealth, poverty, education level, occupational status and welfare benefit receipt, as well as area-level economic indicators and ethnicity, given the strong relationship between ethnicity and lower SES particularly in the USA[22]—as defined by Hillier-Brown et al.[17]

### Objectives

This study has two objectives:
1. To systematically review the quantitative evidence of changes in social security benefit policies on mental health and mental health inequalities.
2. To establish the associational pathways linking social security benefits and mental health.

### Eligibility criteria

Studies will be selected according to the criteria outlined in table 1.

### Interventions

The review will examine a range of social security policies that cover income loss due to poor health and disability, unemployment, low income, single parenthood and old age as well as tax credits and benefits for families with dependent children.

### Outcomes

#### Primary

Studies will be included if one of the outcome measures reported is the number or rate of mental health problems reported. Mental health can be measured both objectively and subjectively. Objective measures include formal diagnoses (eg, anxiety, depression and suicide). Subjective measures include those that are self-reported (eg, self-assessed mental health and well-being scores, days of good/poor mental health) or by validated questionnaires such as the Warwick-Edinburgh Mental Health Scale. For children, outcomes relating to emotional and behavioural aspects of well-being will be included.

Outcomes will be collected as reported. Due to possible variations in disease definitions over time, we will extract definitions of outcomes as reported in individual studies. We will extract outcomes in all data forms (eg, dichotomous, continuous) as reported in the included studies.

Studies including general self-assessed health will not be included unless they also report a mental health component. Outcomes reported for specific population subgroups (eg, by SES) will also be included. There will be no restrictions on the timing of outcomes.

#### Secondary

Secondary outcomes of this study include mental health inequalities, as measured by SES.

**Table 1** Eligibility criteria

| | Inclusion | Exclusion |
|---|---|---|
| Participants (P) | All individuals of any age | |
| Intervention (I) | A national or regional level social security reform or a series of reforms, defined as any government change to: <br>1. Cash benefit levels <br>2. Eligibility/conditionality related to benefit receipt (including changes in eligibility assessment policies) <br>3. Introduction or elimination of a benefits policy | Interventions (ie, not government policies such as employer-funded insurance schemes) <br>Multisectorial policies (eg, Active Labour Market Programmes, broad austerity policies and welfare regimes) <br>Policies not considering cash transfers (eg, in-kind benefits) <br>Parental leave, child-care subsidy policies |
| Comparison (C) | With or without a clearly defined control group | |
| Outcomes (O) | Clearly defined mental health outcomes (eg, depression, anxiety, stress and suicide), and could include subjective measures, such as subjective well-being and life satisfaction, as well as symptoms, events and diagnoses <br>Emotional and behavioural aspects of child well-being | Non-mental health outcomes (eg, general health if there is no separate mental health component) <br>Outcomes related to healthcare utilisation (eg, hospital admissions, antidepressant prescriptions) |
| Study type | Observational studies evaluating change(s) to a specific policy, including: <br>Cohort studies <br>Cross-sectional studies <br>Longitudinal studies <br>Quasi-experimental studies | Descriptive studies reporting mental health outcomes in benefit recipients vs non-recipients or between different recipient groups <br>Randomised controlled trials <br>Qualitative studies <br>Editorials <br>Commentaries <br>Expert opinion articles |
| Study period | Published in the last 40 years (1979–2020) | Literature published before 1979 |
| Setting | High-income countries (as per World Bank definition) | Non-high-income countries |
| Study reporting language | English | |

## Study exclusion

Studies will be excluded if they were published prior to 1979. The year 1979 was selected as a cut-off date because it marks a significant shift in policy from this period onwards going from passive to active welfare in high-income countries. Only studies in high-income countries will be included.

## Study design

Observational study designs will be included if they report mental health outcomes related to a national (or regional) social security policy change and are based on a quantitative method.

## Search strategy

We will examine seven major databases, including MEDLINE, EMBASE, Cumulative Index to Nursing and Allied Health Literature, Applied Social Sciences Index Abstracts, PsycINFO, Scopus and Research Papers in Economics from January 1979 until April 2020.

Search terms were identified from scoping searches and comprise three main concepts related to mental health, social security policy change and a quantitative research method. Although not exhaustive, the search strategy aims to cover a broad range of social security policies, mental health outcomes and quantitative research methods. An information specialist, trained in systematic reviews, as well as a medical librarian were consulted about designing and piloting the search strategy. A sample search strategy adapted for Medline can be found in the online supplementary file 2.

In terms of restrictions, only peer-reviewed English language studies will be included. One exception includes working papers in the field of Economics, as they may comprise a large proportion of relevant literature. Electronic database searches will be supplemented by evaluating the references of literature included in the review as well as the references of other major reviews in the field. Citation searches of included articles will be undertaken using the Science and Social Science Citation Indices. Hand-searching of key journals on health and social policy will also be conducted.

## Study selection

Studies will be imported to EndNote (V.X9)[23] for deduplication and then to an online software programme, Rayyan[24] for screening. One reviewer (JS) will screen all the titles and abstracts, and, given the large numbers of the initial search hits, the second reviewer (ZB) will screen a random sample of 10% of these, with a third reviewer assisting to resolve any disagreements. We will obtain full reports for all titles that appear to meet the inclusion criteria or where there is any uncertainty. If necessary,

additional information will be sought by contacting the study authors. Both reviewers will assess the eligibility of the full texts and appraise the quality of the included studies.

Prior to data extraction, a data extraction form will be created and pilot-tested using a subset of included studies. The form will be modified based on feedback from data extractors in order to improve the usability and appropriateness of the form. The data abstraction framework will be used as a template for recording significant study characteristics of the literature appropriate for inclusion in the review. This information will include details on: study author, year, title, population, mental health outcome measure(s), policy, study design, data source, data years, analytic approach and results. Data will be extracted by two reviewers independently (JS and HB/VA).

### Risk of bias assessment

The quality appraisal will be undertaken by two researchers working independently (JS and HB/VA), using the validity assessment framework developed by Barr and colleagues.[25] It includes nine component rating sections (unit of analysis; comparison approach; sample selection; number of time points of data; response bias; exogeneity of policy exposure, confounding, sample size/power and statistical methods). Based on the performance in terms of the above criteria, each study will be assigned a global score and classed as either 'strong' or 'moderate' or 'weak'. Any disagreements between the researchers will be resolved through a discussion between reviewers, if necessary with a third reviewer (ZB). The quality of the studies will form a part of the narrative synthesis to highlight the variation between studies and to help demonstrate the overall strength of evidence in this study area.

### Data synthesis

Since combining results from individual studies has the advantage of increasing both statistical power and precision in estimating the impact of social security policies, it is beneficial to undertake a meta-analysis of collated studies in circumstances where clinical and methodological homogeneities permit. In the event that sufficient studies of similar construct and outcomes are identified, a meta-analysis of the results will be undertaken. The continuous outcomes relating to mental health will likely be studied by investigation of the policy effect size, through measurement of the standardised mean difference in mental health before and after policy in question. This will be presented in a forest plot, facilitating both visual and statistical comparison. A sensitivity analysis will be undertaken, addressing whether the exclusion of studies with a poor quality rating has an impact on the precision and overall conclusions of the review. This will be done by excluding one paper at a time and exploring the impact on the overall results. Heterogeneity will be quantified by measuring the $I^2$ statistic ($I^2$ statistic with values of 30%–60%, 50%–90% and 75%–100% used to denote moderate, substantial and considerable levels of heterogeneity, respectively[26]). If appropriate, heterogeneity will be further explored by undertaking subgroup analysis. Planned subgroup analyses include those by age, gender and type of policy reform. Publication bias will be assessed using funnel plots, as well as the Egger regression test.[27]

If meta-analysis is not possible, then mental health effects of each policy will be summarised narratively as follows. First, policies will be classified into five broad categories of social security benefits (ie, social assistance; families with children; disability; unemployment and retirement). Policies will be further subdivided by type of reform—as changes to either benefit levels, eligibility or introduction/elimination of policy. Different magnitudes of each policy will be taken into account by assigning them differential weights, using a modified version of the 'direct weighting' approach recommended by the Cochrane group.[28] This will entail assigning each policy different weights depending on: affected population size; change in magnitude in terms of either benefit level, eligibility requirements or the scale of the policy introduced/removed; risk of bias in each study. The direction of effects will be broadly summarised as either more restrictive or generous policy.

In addition, we will present a tabulated summary of key study characteristics that will include a description of policies, outcomes, populations and settings. This will be followed by a clear accompanying descriptive account addressing the robustness of the evidence presented, and the relationships within and between studies included in the review. Since by definition, narrative synthesis is an inherently more subjective process, it is crucial to use a rigorous and transparent approach to the data collected. For this reason, we will use the Economic and Social Research Council framework and guidelines to present the narrative synthesis, as described by Popay and colleagues.[29] More specifically, this will include developing a logic model presenting the potential pathways and mediating factors between changes in social security benefits and mental health (eg, income and employment). As such, the logic model will help guide the structure and interpretation of the findings of the review.

## PATIENT AND PUBLIC INVOLVEMENT

No patients or public are directly involved in this study. The data for systematic review will be collected from previously published studies.

## DISCUSSION
### Strengths and limitations

To our knowledge, this will be the first systematic review to explore the relationship between social security reform and mental health. Building on a number of successful reviews linking social determinants and health,[17 30–32] the wide range of policies included will

provide a comprehensive overview of how social security policies can affect mental health. In our view, taking a comprehensive overview is important because social security or 'welfare' reform usually constitutes a number of policies and thus cannot be reduced to a single intervention. The review will therefore provide valuable information for policy-makers as well as insights for areas for future research. In particular, it will help identify gaps in terms of what types of policies require further investigation. Lastly, by following a systematic and transparent approach and adhering to recommended and validated methods guidelines, we aim to ensure that our findings present a valid representation of the existing evidence.

There are several potential limitations associated with this review. First, including a wide range of policies and mental health outcomes may result in a high heterogeneity between included studies which may limit the statistical power to conduct a meta-analysis. Second, our decision to exclude non-English studies might exclude some relevant policy evaluations, particularly those outside of Europe, Australia or North America. However, due to resource limitations, this is beyond the scope of our review. Further, as with most systematic reviews, our review is susceptible to publication bias whereby only studies with non-null findings are more likely to be published. We aim to minimise this limitation by reviewing the references and citations of the included studies as well as by hand-searching the key journals and including economic working papers, in this way ensuring that the broadest range of relevant literature has been identified. The findings of our review may also be limited by the nature of study designs of the included studies, as they will be observational. We will, however, conduct a rigorous quality assessment of each study and will account for study quality in our narrative synthesis.

## ETHICS AND DISSEMINATION

Since this is a review of secondary data, ethical approval will not be needed. The findings from this review will be disseminated by submission to a peer-reviewed journal. The results will also be presented at a national conference and circulated to the general public and key stakeholder groups using social media.

**Acknowledgements** We would like to acknowledge our peer-reviewers Ben Barr and Patricia O'Campo for their valuable comments.

**Contributors** JS drafted the manuscript and is the guarantor for the review. All authors contributed to the development of the selection criteria, the risk of bias assessment strategy and data extraction criteria. JS developed the search strategy. HB, ZB, VA and CB provided comments and amendments. All authors read, provided feedback and approved the final manuscript.

**Funding** This systematic review is funded by the Economic and Social Research Council, as a part of a PhD project.

**Competing interests** None declared.

**Patient consent for publication** Not required.

**Provenance and peer review** Not commissioned; externally peer reviewed.

**ORCID iD**
Julija Simpson http://orcid.org/0000-0001-8540-5717

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
