## [Reviewer comments · BMJ Open]

ARTICLE DETAILS

TITLE (PROVISIONAL)	Investigating the relationship between changes in social security benefits and mental health: a protocol for a systematic review
AUTHORS	Simpson, Julija; Brown, Heather; Bell, Zoe; Albani, Viviana; Bamba, Clare

VERSION 1 - REVIEW

REVIEWER	Patricia O'Campo University of Toronto, Canada I have, in the past, had Claire Bamba, as a co-author on one of my papers.
REVIEW RETURNED	23-Dec-2019

GENERAL COMMENTS	This is a thorough protocol for an ambitious systematic review of social protection policies by a team that has been engaged in this area of research for many years. More information on the policy exposure should be provided. While the authors mention a full range of policies of interest, yet it is the policy reform piece that needs further articulation. What qualifies as a policy reform for the purpose of this review? The reader would benefit from knowing more about how small (e.g., change in eligibility) or how large (e.g., elimination of a policy) a change needs to be for the study to be included in the review. How might the magnitude/nature of the policy reform be considered in the analyses? While one strength of the study is that multiple social protection policies are being examined, the authors might provide a clearer explanation of the benefit of looking at so many policies together in a single systematic review. What are the gains over reviewing one or a few related social protection policies? About the "inequality" outcome, the research question suggests that inequalities alone is being examined (i.e., it says mental health and inequalities) but the Methods talks about mental health inequalities. The inconsistency should be clarified and also the definition of inequality should be further elaborated upon so that it is clear what outcomes are of interest here. The analysis section would benefit from greater attention. How will different subpopulations be handled in the analyses? For example, how might the impact on younger vs older workers mental health in response to changes to (un)employment related benefits be
--

	considered in the analyses? Same could be asked about differential impacts on men vs women or single adults versus single headed households. The authors do not mention this aspect of the analyses. How will context be taken into account in the analyses such as policy contexts (e.g., welfare state).
--	--

REVIEWER	Ben Barr University of Liverpool
REVIEW RETURNED	06-Jan-2020

GENERAL COMMENTS	The paper presents a systematic review proposal on an important topic - investigating the mental health impact of social security reforms. There are however are however few changes that would help clarify the review, improve the study and reduce bias and enable the meaningful interpretation of results. Introduction. The introduction would benefit from including a discussion of other relevant systematic reviews. There have been systematic reviews of the health effect of policy changes that influence income - these include social security changes - for example Cooper K, Stewart K. Does Money Affect Children's Outcomes?: A Systematic Review. Joseph Rowntree Foundation 2013. 1 Cooper K, Stewart K. Does money in adulthood affect adult outcomes? York: : Joseph Rowntree Foundation 2015. http://www.jrf.org.uk/publications/does-money-adulthood-affect-adult-outcomes (accessed 30 Jul 2015). There have also been systematic reviews of cash transfer schemes in low / middle income countries that would meet their definition of social security policies. Although these are focused on a different set of countries - there is actually some overlap - e.g Uruguay which is defined as a high income country based on their definition http://worldpopulationreview.com/countries/high-income-countries/ but as a middle income country in other reviews, possibly because countries can move in or out of this group over time. It would be useful to discuss this SR in the context of these other SR. Definiton of "social security policy". The definition of a "social security policy" is insufficiently clear for the purposes of the review. The authors refer to these as: "National or regional level policy relating to changes in social security benefits" that "cover income loss due to poor health and disability, unemployment, low income, single parenthood and old age as well as tax credits and benefits for families with dependent children." The definition will need to clarify whether this just includes policies that involve cash transfers, or would it also include those that involve other in-kind benefit e.g food stamps, healthy start, crisis loans etc. Is the SR limited to government policies? Many countries have mandatory employer funded insurance schemes covering workplace
---

disability - called worker compensation schemes (e.g US/ Canada/ Australia). Whilst these carry out similar function to incapacity benefits in the UK they are not exactly governmental, although covered by national legislation. Presumably private disability insurance schemes are not covered - but the lines between government / non-governmental / and private are blurred in some country contexts. This will need to be clarified in the protocol.

In addition the protocol could clarify which changes to these policies are included. Social security is a huge policy area with a potentially infinite number of different policy changes. The major changes in high income countries in recent years have been (1) changes to eligibility (2) changes to assessment processes (3) changes to the administration (e.g subcontracting parts of the process to private contractors) (3) changes to conditionality often linked to ALMPs (4) changes to benefit level (e.g income replacement rates)... but there are many other changes. There are potentially multiple complex pathways through which these could effect mental health - e.g through income effects, through stress uncertainty / stigma of the administrative process, through employment effects etc. The protocol should clarify which changes it proposes to include in the SR - but also as recommended by Tugwell et al - include a logic model that shows the hypothesised relation between these changes and mental health outcomes. This will greatly support the interpretation of the findings and enable the review to be structured so it produces relevant evidence.

Study design.

In previous systematic reviews of social security intervention we have conducted - the majority of studies identified are quasi-experimental/ econometric evaluations of natural experiments - generally using one of the following methods: ITS, Difference in Differences, Regression discontinuity, instrumental variables or other panel regression techniques. The protocol would benefit from being explicit about the exclusion of some of these study designs. For example - Regression discontinuity, and instrumental variables methods often use idiosyncrasies in eligibility criteria and thresholds to identify causal effects and are relatively common in this area of research - these would however be excluded based on the proposed study design criteria as they will generally not involve a before and after comparison. If this is the intention - it would be worth being explicit about this and giving a justification.

Whilst the number of randomised controlled trials of social security policies is small, they do exist - it is not clear why the study type in table 1 only includes Non-randomised or uncontrolled trials. I cannot see any justification of not including RCTs if they exist.

Search strategy.

Based on previous reviews of social security policy we have conducted - the majority of the research is in the economics literature. The publication process in the field of economics generally involves initial publication of working papers - it may then be several years before these are then published in a journal. Without including economic databases in their search stagey or searches of databases of economic working papers - the review will probably miss a large proportion of the relevant literature. I would suggest the authors include a search of RePEc (Research Papers in Economics) - <http://repec.org/> which can be searched using the terms they

	describe. Risk of bias assessment. Without some adaption the Effective Public Health Project tool is unlikely to provide much quality discrimination between the type of studies this review is likely to identify - quasi-experimental/ econometric evaluations of natural experiments. This is because the EPHP tool is developed for traditional clinical/ epidemiological studies - RCTs, cohort etc, and not evaluations of natural experiments. The main risk of bias in these types of studies is unobserved confounding, and the methods used e.g DiD, synthetic controls, fixed effects regression etc aim to reduce this risk of bias - however even though these methods if implemented appropriately can reduce the risk of bias from unobserved confounding - this would not be picked up through the EPHP tool. Previously we have developed our own tool for natural experiment type studies as existing tools were not appropriate (see https://livrepository.liverpool.ac.uk/3034626/) Outcomes. The protocol should clarify if studies investigating outcomes related to health care utilisation are going to be included - e.g - antidepressant prescribing rates, hospital attendances for self harm etc? For the secondary health inequalities outcome - they should clarify whether by HI outcomes - they mean differential effects by SES, or differential exposure to policy by SES but the same effect on the exposed - or both. Meta-analysis: If meta analysis is to be carried out this should include assessment of publication bias - e.,g using funnel plots etc. References. 1 Pega F, Liu SY, Walter S, et al. Unconditional cash transfers for reducing poverty and vulnerabilities: effect on use of health services and health outcomes in low- and middle-income countries. Cochrane Database Syst Rev 2017;2017. doi:10.1002/14651858.CD011135.pub2 1 Cooper K, Stewart K. Does Money Affect Children's Outcomes?: A Systematic Review. Joseph Rowntree Foundation 2013. 1 Cooper K, Stewart K. Does money in adulthood affect adult outcomes? York: : Joseph Rowntree Foundation 2015. http://www.jrf.org.uk/publications/does-money-adulthood-affect-adult-outcomes (accessed 30 Jul 2015).
--	---

VERSION 1 – AUTHOR RESPONSE

	Comments	Response
	Reviewer #1	
1.	More information on the policy exposure should be provided. While the authors mention a full range of policies of interest, yet it is the policy reform piece that needs further	Thank you for this point. We have clarified the definition of social security policy reform in Table 1 (p.4) as follows: Intervention: A national or regional level social

	articulation. What qualifies as a policy reform for the purpose of this review? The reader would benefit from knowing more about how small (e.g., change in eligibility) or how large (e.g., elimination of a policy) a change needs to be for the study to be included in the review.	security reform, defined as any change to:  1) Benefit levels; 2) Eligibility/conditionality related to benefit receipt (including changes in eligibility assessment policies); 3) Introduction or elimination of an existing benefit policy.
2.	How might the magnitude/nature of the policy reform be considered in the analyses?	We have added in the Data Synthesis section (p.7-8) that policies will be classified into five key domains:  1. Social Assistance policies (e.g. income support, housing benefit); 2. Policies aimed at families with children (e.g. tax credits, child benefit); 3. Disability benefits; 4. Unemployment benefits; 5. Retirement benefits. These policies will be further sub-divided as change to either:  1. Benefit level; 2. Eligibility; 3. Introduction/elimination of policy. We agree that it is important to take account of the differing magnitudes of the different policies. Thus, we have added in p.8 that we will apply a modified version of the 'direct weighting' approach, recommended by the Cochrane group (1). This will entail assigning each policy different weights depending on:  - affected population size; - change in magnitude in terms of either benefit level, eligibility requirements or the scale of the policy introduced/removed; - risk of bias in each study. The direction of effects will be broadly summarised as either more restrictive or generous policy.
3.	While one strength of the study is that multiple social protection policies are being examined, the authors might provide a clearer explanation of the benefit of looking at so many policies together in a single systematic review. What are the gains over reviewing one or a few related social protection policies?	We agree that the benefits of such a broad systematic review could be better articulated. We have expanded on this in the Discussion section (p.8) by adding: "In particular, it will help identify gaps in terms of what types of policies require further investigation". We have also noted a number of successful reviews taking broad approaches – reviews upon which we will

		build upon (e.g. McAllister et al, 2018 (2); Naik et al, 2019 (3)). Furthermore, the lack of evidence in this area has been emphasised in the Introduction section (p.3). In addition, we have noted that we are interested in the effects of welfare reform as a whole. Our view is that welfare reform cannot be reduced to single interventions but constitutes a package. We will however look at different types of policies in our meta-analysis or narrative synthesis (i.e. Changes to benefit levels; Eligibility/conditionality related to benefit receipt; Introduction or elimination of an existing benefit policy).
4.	About the “inequality” outcome, the research question suggests that inequalities alone is being examined (i.e., it says mental health and inequalities) but the Methods talks about mental health inequalities. The inconsistency should be clarified and also the definition of inequality should be further elaborated upon so that it is clear what outcomes are of interest here.	Apologies for the inconsistency. The focus of the review is on mental health inequalities. We have clarified the terminology and ensured that it is used consistently throughout. We have defined mental health inequalities as ‘differences in mental health effects by commonly used indicators of socio-economic status such as individual income, wealth, poverty, education level, occupational status, welfare benefit receipt; as well as area-level economic indicators and ethnicity given the strong relationship between ethnicity and lower SES particularly in the USA (4) – as defined by Hiller-Brown et al’ (5)) (p.4).
5.	The analysis section would benefit from greater attention. How will different subpopulations be handled in the analyses? For example, how might the impact on younger vs older workers mental health in response to changes to (un)employment related benefits be considered in the analyses? Same could be asked about differential impacts on men vs women or single adults versus single headed households. The authors do not mention this aspect of the analyses.	Thank you. We have added that, if meta-analysis is possible, we will do sub-group analysis (e.g. by age, gender, type of policy reform). If not, then the narrative synthesis will examine these additional stratifications (p.7).
6.	How will context be taken into account in the analyses such as policy contexts (e.g., welfare state).	We will take into account different welfare state contexts in the Discussion section. Specifically, we will be noting that reforms that have effects in one context cannot be applied into others and expect the same results – as was acknowledged by Hiller-Brown et al (5).
	Reviewer #2	
7.	The introduction would benefit from including a discussion of other relevant systematic reviews. There have been systematic reviews of the health effect of policy changes that influence income - these include social	Thank you for the suggestion. We have incorporated the additional references in the Introduction section to strengthen the background of the review (p.3).

	security changes - for example Cooper K, Stewart K. Does Money Affect Children's Outcomes?: A Systematic Review. Joseph Rowntree Foundation 2013. Cooper K, Stewart K. Does money in adulthood affect adult outcomes? York: : Joseph Rowntree Foundation 2015. http://www.jrf.org.uk/publications/does-money-adulthood-affect-adult-outcomes (accessed 30 Jul 2015).	
8.	There have also been systematic reviews of cash transfer schemes in low / middle income countries that would meet their definition of social security policies. Although these are focused on a different set of countries - there is actually some overlap - e.g Uruguay which is defined as a high income country based on their definition http://worldpopulationreview.com/countries/high-income-countries/ but as a middle income country in other reviews, possibly because countries can move in or out of this group over time. It would be useful to discuss this SR in the context of these other SR.	Although we are not entirely certain which systematic review the reviewer is referring to, we have included two examples of systematic reviews (Bastagli, 2019 (6); Lagarde, 2009 (7)) in low- and middle-income countries in the Introduction (p.3)
9.	The definition of a "social security policy" is insufficiently clear for the purposes of the review. The authors refer to these as: "National or regional level policy relating to changes in social security benefits" that "cover income loss due to poor health and disability, unemployment, low income, single parenthood and old age as well as tax credits and benefits for families with dependent children." The definition will need to clarify whether this just includes policies that involve cash transfers, or would it also include those that involve other in-kind benefit e.g food stamps, healthy start, crisis loans etc. Is the SR limited to government policies? Many countries have mandatory employer funded insurance schemes covering workplace disability - called worker compensation schemes (e.g US/ Canada/ Australia). Whilst these carry out similar function to incapacity benefits in the UK they are not exactly governmental, although covered by national legislation. Presumably private disability insurance schemes are not covered - but the	We thank you for the specific points for clarification. To be clear, we are focussing on government transfers only. We have added this and other clarification points to Table 1 (p.4). Please also see our response to comment no. 1.

	lines between government / non-governmental / and private are blurred in some country contexts. This will need to be clarified in the protocol.	
10.	In addition the protocol could clarify which changes to these policies are included. Social security is a huge policy area with a potentially infinite number of different policy changes. The major changes in high income countries in recent years have been (1) changes to eligibility (2) changes to assessment processes (3) changes to the administration (e.g subcontracting parts of the process to private contractors) (3) changes to conditionality often linked to ALMPs (4) changes to benefit level (e.g income replacement rates)... but there are many other changes. There are potentially multiple complex pathways through which these could effect mental health - e.g through income effects, through stress uncertainty / stigma of the administrative process, through employment effects etc.	Please see our response to comment no.1. We have clarified the definition in Table 1 (p.4)
11.	The protocol should clarify which changes it proposes to include in the SR - but also as recommended by Tugwell et al - include a logic model that shows the hypothesised relation between these changes and mental health outcomes. This will greatly support the interpretation of the findings and enable the review to be structured so it produces relevant evidence.	Thank you for the suggestion. We have added the development of a logic model to the Data Synthesis section (p.8)
12.	Study design. In previous systematic reviews of social security intervention we have conducted - the majority of studies identified are quasi-experimental/ econometric evaluations of natural experiments - generally using one of the following methods: ITS, Difference in Differences, Regression discontinuity, instrumental variables or other panel regression techniques. The protocol would benefit from being explicit about the exclusion of some of these study designs. For example - Regression discontinuity, and instrumental variables methods often use idiosyncrasies in eligibility criteria and thresholds to identify causal effects and are relatively common in this area of research - these would however be excluded based on	Since we are including a broad range of policy changes in the review (i.e. changes in benefit levels; eligibility/conditionality related to benefit receipt; and introduction or elimination of an existing benefit policy), a number of different econometric techniques may be applied and we feel that it may be too restrictive to exclude studies based solely on their data analysis method. However, in addition to the other non-experimental designs, we have explicitly added 'quasi-experimental' designs to Table 1 (p.5).

	the proposed study design criteria as they will generally not involve a before and after comparison. If this is the intention - it would be worth being explicit about this and giving a justification.	
13.	Whilst the number of randomised controlled trials of social security policies is small, they do exist - it is not clear why the study type in table 1 only includes Non-randomised or uncontrolled trials. I cannot see any justification of not including RCTs if they exist.	We have chosen to exclude randomised studies as they typically include policy experiments rather than national/regional policy changes. For example, they include evaluations of specific initiatives of Active Labour Market Programmes which is one of our exclusion criteria and have been reviewed before (e.g. Gibson, 2018 (8)). We have clarified that we include only 'observational' study designs in Table 1 (p.5).
14.	Search strategy. Based on previous reviews of social security policy we have conducted - the majority of the research is in the economics literature. The publication process in the field of economics generally involves initial publication of working papers - it may then be several years before these are then published in a journal. Without including economic databases in their search stage or searches of databases of economic working papers - the review will probably miss a large proportion of the relevant literature. I would suggest the authors include a search of RePEc (Research Papers in Economics) - http://repec.org/ which can be searched using the terms they describe.	Thank you for suggesting this valuable and comprehensive source of economic literature. We have added this to our search strategy (p.6) and removed the limitation of not searching grey literature.
15.	Risk of bias assessment. Without some adaption the Effective Public Health Project tool is unlikely to provide much quality discrimination between the type of studies this review is likely to identify - quasi-experimental/ econometric evaluations of natural experiments. This is because the EPHP tool is developed for traditional clinical/ epidemiological studies - RCTs, cohort etc, and not evaluations of natural experiments. The main risk of bias in these types of studies is unobserved confounding, and the methods used e.g DiD, synthetic controls, fixed effects regression etc aim to reduce this risk of bias - however even though these methods if implemented appropriately can reduce the risk of bias from unobserved confounding - this would not be picked up through the EPHP tool. Previously we have developed our own tool for natural experiment type	Thank you for suggesting a more targeted and appropriate tool. We agree that that EPHP may not be sensitive enough for appraising natural experiments (which, we expect, will comprise the majority of the included papers). We have changed the quality assessment tool to the Validity Assessment framework in the Risk of Bias Assessment section (p.7).

	studies as existing tools were not appropriate (see https://livrepository.liverpool.ac.uk/3034626/)	
16.	Outcomes. The protocol should clarify if studies investigating outcomes related to health care utilisation are going to be included - e.g - antidepressant prescribing rates, hospital attendances for self-harm etc?	We will be excluding healthcare utilisation outcomes. This has been added as an exclusion criterion in Table 1 (p.5).
17.	For the secondary health inequalities outcome - they should clarify whether by HI outcomes - they mean differential effects by SES, or differential exposure to policy by SES but the same effect on the exposed - or both.	We have included a more specific definition of mental health inequalities in the Systematic Review Questions section (p.4) - as in our response to comment no. 4.
18.	Meta-analysis: If meta analysis is to be carried out this should include assessment of publication bias - e.g using funnel plots etc.	Thank you for the suggestion – this has been added to the Data Synthesis section (p.7).

References

1. Deeks J, Higgins JP, Altman D. Analysing data and undertaking meta-analyses. In: Higgins J, Thomas J, Chandler J, Cumpston M, Li T, Page M, et al., editors. *Cochrane Handbook for Systematic Reviews of Interventions* version 602019.
2. McAllister A, Fritzell S, Almroth M, Harber-Aschan L, Larsson S, Burstrom B. How do macro-level structural determinants affect inequalities in mental health? - a systematic review of the literature. *Int J Equity Health*. 2018;17(1):180.
3. Naik Y, Baker P, Ismail SA, Tillmann T, Bash K, Quantz D, et al. Going upstream - an umbrella review of the macroeconomic determinants of health and health inequalities. *BMC public health*. 2019;19(1):1678.
4. Williams DR, Collins C. US socioeconomic and racial differences in health: patterns and explanations. *Annual review of sociology*. 1995;21(1):349-86.
5. Hillier-Brown F, Thomson K, McGowan V, Cairns J, Eikemo TA, Gil-González D, et al. The effects of social protection policies on health inequalities: evidence from systematic reviews. *Scandinavian journal of public health*. 2019;47(6):655-65.
6. Bastagli F, Hagen-Zanker J, Harman L, Barca V, Sturge G, Schmidt T. The impact of cash transfers: A review of the evidence from low-and middle-income countries. *Journal of Social Policy*. 2019;48(3):569-94.
7. Lagarde M, Haines A, Palmer N. The impact of conditional cash transfers on health outcomes and use of health services in low and middle income countries. *The Cochrane database of systematic reviews*. 2009(4):Cd008137.
8. Gibson M, Thomson H, Banas K, Lutje V, McKee MJ, Martin SP, et al. Welfare-to-work interventions and their effects on the mental and physical health of lone parents and their children. *Cochrane Database of Systematic Reviews*. 2017(8).

VERSION 2 – REVIEW

REVIEWER	Ben Barr University of Liverpool, UK
REVIEW RETURNED	02-Mar-2020

GENERAL COMMENTS	The Authors have adressed all the points I previously made, except one, point number 12. I probably wasn't clear enough and I think they may have misunderstood my point. I agree that " it may be too restrictive to exclude studies based solely on their data analysis method." The point that I was trying to make is that their current criteria would do exactly that - since they say there are only including "Before and after (with or without control groups), " this would exclude many RD or IV studies that would often not include before and after measurements and may be cross sectional. If that is the intention that is fine but needs to be jutsified. If it is not the intention the Comparison inclusion criteria needs to be changed to also include cross sectional studies using RD or IV methods. Thanks Ben Barr
--

VERSION 2 – AUTHOR RESPONSE

Reviewer's Comment

"I agree that " it may be too restrictive to exclude studies based solely on their data analysis method." The point that I was trying to make is that their current criteria would do exactly that - since they say there are only including "Before and after (with or without control groups), " this would exclude many RD or IV studies that would often not include before and after measurements and may be cross sectional. If that is the intention that is fine but needs to be jutsified. If it is not the intention the Comparison inclusion criteria needs to be changed to also include cross sectional studies using RD or IV methods."

Our Response

First, thank you very much for all the helpful comments and suggestions – they have definitely helped us clarify and improve the protocol. Regarding the above point, apologies for the misunderstanding. We agree with your suggestion to broaden the inclusion criteria which would include the RD or IV studies. We have expanded the inclusion criteria to include the majority of observational designs as long as they evaluate a change in policy. We have removed the 'before and after' restriction.

The Study Design eligibility criteria in Table 1 (p.5) now are as follows:

Inclusion Criteria: Observational studies evaluating change(s) to a specific policy (with or without a clearly defined control group), including: Cohort studies, Cross-sectional studies, Longitudinal studies and Quasi-experimental studies.

Exclusion Criteria: Descriptive studies reporting mental health outcomes in benefit recipients vs non recipients or between different recipient groups, Randomised controlled trials, Qualitative studies, Editorials, Commentaries, Expert opinion articles.